# Exploring strategies for management of in-hospital stroke in Sweden: A qualitative study

Ilan Ben-Shabat[1]*, Kristina Lindvall[2], Jonatan Salzer[1]

1 Department of Clinical Sciences, Neurosciences, Umeå University, Umeå, Sweden, 2 Department of Epidemiology and Global Health, Umeå University, Umeå, Sweden

* ilan.ben-shabat@umu.se

## Abstract

### Background

Patients with in-hospital stroke (IHS) are discovered and treated with delays compared to community-onset stroke. This qualitative study explores current routines and clinical practices for IHS in Sweden, aiming to uncover factors influencing management and propose areas for future research and development.

### Methods

Six physicians in charge of stroke alerts at Swedish hospitals were individually interviewed in video calls. Informants were selected from The Swedish Stroke Register, based on the hospital-specific median processing time for delivering thrombolysis or thrombectomy to IHS patients, stratified by hospital size. Transcribed interviews were analysed using reflexive thematic analysis.

### Results

Three main themes were developed. The first emphasized the crucial step of discovering IHS and outlined possible workflow pathways, including defining the "key player" with stroke expertise and mandate to proceed with the stroke alert to immediate radiology. Subsequent themes addressed obstacles to optimal practice and suggested clear guidelines for contacting the "key player" to reduce delays, as well as offering IHS education to hospital staff.

### Conclusions

This study identified differences in workflows for IHS management across the six included sites. A "key player" emerged as a common denominator, who was called as the initiation of the stroke alert and had mandate to proceed with the alert to immediate radiology. Clear guidelines for contacting the "key player" and increased education about IHS were suggested as possible ways to mitigate delays to activate the stroke alert.

**Editor:** Sonu Bhaskar, Global Health Neurology Lab / NSW Brain Clot Bank, NSW Health Pathology / Liverpool Hospital and South West Sydney Local Health District / Neurovascular Imaging Lab, Clinical Sciences Stream, Ingham Institute, AUSTRALIA

**Data Availability Statement:** All relevant data are within the paper and its Supporting Information files.

**Funding:** IB, 250.000 SEK, Region Norrbotten, https://www.norrbotten.se JS, 100.000 SEK, VISARE NORR Fund, Northern Country Councils Regional Federation, https://www.norrasjukvardsregionforbundet.se/halso-och-sjukvard/fou/visare-norr/ This study was conducted without involvement of the funders.

**Competing interests:** The authors have declared that no competing interests exist.

## Introduction

In Sweden, in-hospital stroke (IHS) accounts for 5.4% of all strokes [1], with international reports ranging from 2.2–16.6% [2–6]. This subset of stroke patients experiences stroke onset while already hospitalized for an existing illness or trauma, with 47% having recently undergone an invasive procedure or surgery [1]. Patients who experience a stroke while hospitalized have been shown to face delays in symptom identification [7], and studies report differences in care for IHS patients compared to those with community-onset stroke. These include lower rates of revascularization therapy [6], delayed treatment delivery for eligible patients [2, 6], and lower adherence to care guidelines [2]. Outcomes for IHS patients are generally unfavorable, with higher rates of discharge to care facilities and increased post-stroke mortality [2, 6]. Guidelines emphasize that prompt initiation of a stroke chain for rapid imaging and evaluation is essential for timely decision-making regarding reperfusion therapy [8]. The implementation of both pre-hospital and in-hospital system interventions for community-onset stroke (COS) patients has proven effective in increasing revascularization therapy rates for ischemic stroke patients, reducing delays in therapy administration, and ensuring rapid and appropriate treatment for patients with hemorrhagic stroke [9]. These interventions have also contributed to reductions in mortality and morbidity [10, 11]. In 2022 the American Heart Association (AHA) released a statement specifically aimed at improving IHS management [12]. This included educating hospital staff about stroke symptoms and treatment, implementing a standardized assessment method for stroke symptoms, and defining an in-hospital stroke alert activation process, including a dedicated team to respond to these alerts. Implementing structured IHS protocols has proven effective in single-center intervention studies by reducing the time from "last seen well" to assesment and imaging [13–17], as well as increasing the rate of administered revascularisation treatment [14, 17, 18] between pre- and post-intervention periods. The pre-intervention periods in these studies identified several barriers to effective management: lack of knowledge about stroke identification [13, 19] and the need for timely evaluation [13], poorly defined criteria for stroke alert activation [17, 19], uncertainty about roles [17], poor communication [19], and insufficient knowledge about the rapid-response team [13, 17]. Additionally, a lack of resources has previously been cited as a barrier to implementing best-practice stroke management [20].

Currently, there is no national guideline for IHS management in Sweden. Healthcare is largely governed by 21 independent counties, leading to the existence of many local guidelines. To our knowledge, there is a lack of studies describing the IHS workflow beyond single-center experiences, with limited focus on understanding the factors influencing IHS management. This study aims to describe the current routines and clinical practices for IHS management in Sweden, identify factors affecting IHS management, and highlight potential areas for future research and development.

## Methods

### Design, settings and informants

We conducted a qualitative study using semi-structured interviews to gather information about IHS management from physicians responsible for stroke alerts at Swedish hospitals. Informants were identified through the Swedish Stroke Register (*Riksstroke*). A purposive sampling method was used to select the study informants. This process involved extracting aggregated data on lead times for in-hospital ischemic stroke patients who received thrombolytic therapy or endovascular thrombectomy between 2010 and 2019 from the Swedish Stroke Register. Sites were classified based on hospital type and available treatments: comprehensive

stroke centers with on-site thrombectomy (n = 6), specialized non-university hospitals or university hospitals without on-site thrombectomy (n = 28), and community hospitals (n = 41). Inclusion criteria required each hospital to have treated a minimum of 10 patients with any form of revascularization therapy between 2010 and 2019.

For informant selection, hospitals without on-site thrombectomy were chosen based on median onset-to-needle time, while comprehensive stroke centers were selected based on median onset-to-thrombectomy time. Hospitals with the highest and lowest median times in each group were invited to participate. Informants were initially contacted via email by IB, followed by a reminder after two weeks. If no respinse was received after one month, two additional phone contact attempts were made. In cases of refusal or non-response, the hospital with the second-highest or second-lowest median time in the same category was contacted. The planned inclusion goal was two sites per group, for a total of six sites. Recruitment of informants took place continuously between September 2021 and March 2022.

### Data collection

A semi-structured interview guide (Table 1) was created by the authors, containing questions aimed at exploring the IHS workflow, including its processes, influencing factors, and potential areas for improvement. The guide was shared with informants in advance to allow time for reflection.

Semi-structured individual interviews took place via video calls between October 2021 and March 2022, conducted by the first author in Swedish. Informants were encouraged to freely discuss the topics outlined in the interview guide. Regular topic summaries were provided during the interviews to clarify the information. The interviews, which averaged 35 minutes (range 22–46), were recorded and transcribed verbatim by the first author. Transcriptions were completed following each interview and informed subsequent interviews, allowing for reflections on preliminary results and potential follow-up questions. However, the interview guide itself remained unchanged throughout the study.

The initial sample of six sites was deemed sufficient based on to information power criteria [21]. This was primarily due to the specific aim of the study, the highly relevant characteristics of the informants, and the richness of the dialogues between the first author and the informants.

### Data analysis

We used reflexive thematic analysis (RTA) [22], following the six-phase process, while allowing flexibility to move back and forth through the phases as necessary. JS (consultant neurologist) and IB (neurology resident) are both medical doctors with experience in acute stroke workflows, while KL is a public health researcher with expertise in qualitative data analysis.

Familiarization with the data began during collection, as IB conducted, manually transcribed, and shared the interviews with JS and KL. After data collection was complete, the authors held a group session to discuss the content and refine the analytical process. Each author independently generated initial codes for the first interview, and these codes were discussed during a subsequent group session. Codes that were mutually agreed upon were retained, while conflicting codes were discussed, modified, and ultimately agreed upon. IB then coded subsequent interviews, with JS and KL reviewing them and suggesting any additional codes or possible code merging. In a subsequent group session, all codes were examined and categorized to generate initial themes. Further analysis, conducted both individually and in group sessions, continued throughout the report writing process, during which some themes were refined, merged, or renamed. This process ultimately led to the formulation of

**Table 1. Interview guide.**

**Interview Guide**

The interview is semi-structured; themes and questions outside of those listed below are expected to arise during the interview. The questions listed under "if yes" and "if no," as well as "if good" and "if not good," serve as examples of follow-up questions.

**Interview Guide: Stroke Onset in Hospital Compared to Onset in the Community**

*Start by briefly explaining the study, the purpose of the interview, and that the information will be handled confidentially.*

**1. Background and Routines**

a. Brief summary of the interviewee's background:

• Professional title

• Specialist training

• How long have you been responsible for the stroke alert protocol?

b. Is there a routine at your hospital for how to act if an inpatient is suspected of having a stroke?

(If YES to b)

c. Can you describe the various steps included in the routine?

d. How are these patients managed in practice? Is the routine followed?

e. Do you think there is a general awareness of the routine in the hospital? How would you describe the level of awareness among staff in: (*where do you perceive awareness to be lacking or particularly good*)

• Stroke wards?

• OR/post-op/ICU?

• Other departments in the hospital (Internal medicine, Surgery, Orthopedics, etc.)?

f. Are new colleagues informed about the procedure upon employment/internship/temporary work? How are they informed? If they are not informed, what do you think the reason could be?

(If NO to b)

g. How are these patients (inpatients with suspected stroke) managed in practice?

**2. Evaluation**

a. How do you perceive that the acute care chain for this patient group functions in your department? Does it work well? Is there anything that does not work as well?

(If GOOD in a)

b. What is the reason that the acute care chain at your hospital works well?

(If NOT GOOD in a)

c. What is the reason that the acute care chain at your hospital does not work well? Do you perceive that there is any step in the acute care chain where problems occur?

(All)

d. Have you or anyone you know received feedback from the staff who have managed stroke cases with in-hospital onset on how it went? If yes, what feedback have you received?

**3. Improvement Suggestions**

a. Are there are any critical points in the care chain where delays in the care process could occur? If yes, how could these be overcome?

b. Is there any part of the care chain in your department that you would like to improve? How? Why?

c. Have any improvement suggestions been received from the staff in the hospital?

**4. Summary**

*Summarize what was discussed during the interview.*

a. Do you have anything else you would like to add?

Translated version of the interview guide that was sent to the participants in advance and used by IB during the interviews. ICU: intensive care unit, OR: operating room

three themes and ten subthemes. During the post-submission revision process, three subthemes were renamed, and one was moved to a different theme. Transcripts were analysed in their original language, with selected quotes translated into English by IB. Informants were not asked to review or correct the transcripts or results.

## Ethics

This study was approved by the Swedish Ethical Review Authority (Dnr 2020–04857) and adhered to the "Standards for Reporting Qualitative Research" (SRQR). Informants received written information and provided written informed consent. Prior to the interviews, they were also given oral information about confidentiality and the option to withdraw consent at any time. The transcripts were pseudonymized.

**Table 2. Themes and sub-themes.**

| Themes | Sub-themes |
|---|---|
| 1. Discovery and alert | • *Discovery is crucial*<br>• *The "key player"*<br>• *The IHS alert* |
| 2. Obstacles to optimal practise | • *The unknown IHS routine*<br>• *Lack of confidence*<br>• *A robust general stroke alert facilitates IHS management*<br>• *Resignation* |
| 3. Knowledge and experience | • *Varying levels of knowledge about IHS*<br>• *Knowledge and experience improves IHS management*<br>• *Education is requested* |

Themes and sub-themes developed from the interviews. IHS: in-hospital stroke

## Results

Of in total 75 sites in Sweden, 40 met the revascularisation therapy volume criteria. Seven sites were contacted and one site was unreachable. Of the six participating physicians, four were neurologists (4–12 years of stroke alert responsibility) and two were internists (1 and 5 years of stroke alert responsibility). Sites treated in median 11.5 patients between 2010 and 2019 (range 10–28). The median ONT and DNT were 92.5 and 69.5 minutes, respectively, for IHS and 119 and 40 minutes for COS.

Three themes were developed (Table 2): "Discovery and alert", "Obstacles to optimal practice" and "Education and knowledge".

### 1. Discovery and alert

The first theme addresses aspects of current practise, focusing on patient discovery, the subsequent steps following the identification of a suspected IHS patient, and the role of the "key player" in IHS patient management. The process of initiating of the IHS alert is also outlined. Fig 1 summarizes potential management pathways at the six included sites.

**Discovery is crucial.**   Informants unanimously emphasized identifying the IHS patient as crucial in the stroke care chain and agreed that delayed discovery was the primary cause of treatment delays.

"…that is the critical part, if no one realises that the patient had a stroke on the ward. That would certainly result in a huge delay." (Inf1)

They described scenarios contributing to delayed patient discovery, including patients waking up with stroke symptoms, experiencing symptoms post-anesthesia, or becoming symptomatic between staff rounds.

"*Some patients are identified after waking up from anaesthesia… or the patient is not under constant observation at night and symptoms are observed in the morning." (Inf4)*

Some informants recounted incidents where staff were hesitant about whether symptoms indicated a stroke, potentially impeding the alert process. They explained that the individual who identified the patient could consult with a nearby colleague to discuss their findings before informing the ward physician, who would then assess the patient to determine if a stroke was suspected and initiate the stroke alert by calling the "key player".

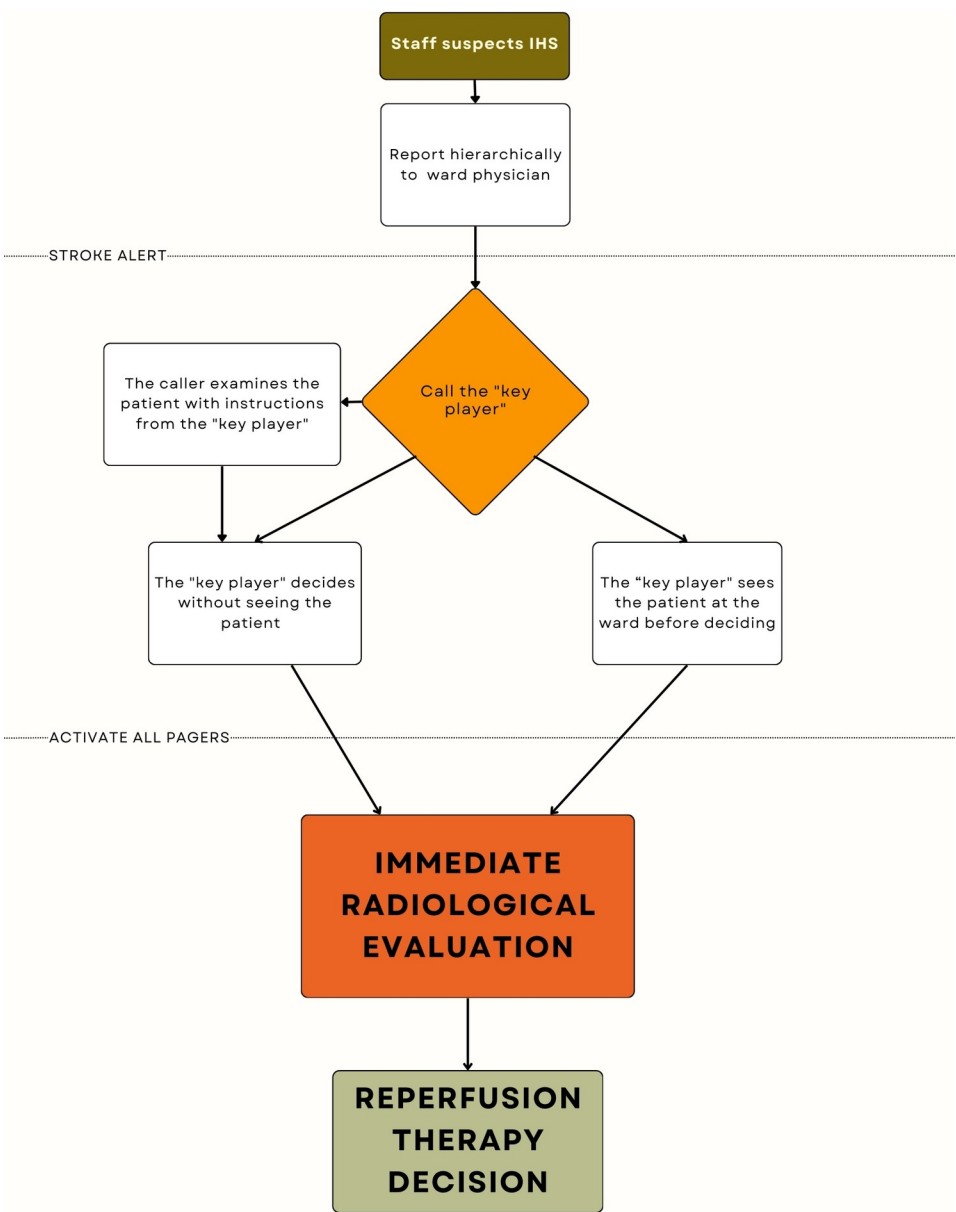

**Fig 1. Potential IHS workflows.** Potential workflows from patient discovery to stroke alert at the six included sites. IHS = in-hospital stroke.

*"I felt as if they had thought about calling me for some time before they actually did." (Inf6)*

*If a nurse assistant suspects an IHS, they would inform the nurse, who, in turn, pages the doctor... The doctor then assesses the patient to determine if a stroke alert is warranted."(Inf6)*

**The "key player".** All informants described the first step in initiating a stroke alert as a telephone call, typically made by the ward physician, to a person with stroke expertise who has the authority to proceed with the stroke alert to immediate radiology. This individual, referred to as the "key player", could be the on-call neurologist or internist, a stroke-specialized nurse, or an emergency physician. At sites with a written IHS protocol, the appropriate person to

contact when suspecting a stroke in a hospitalized patient was clearly defined. At the one site without a written routine, most staff reported calling the on-call internist, as it was considered the most reasonable approach in that context.

When the "key player" was notified of a suspected IHS patient, various scenarios were described. Some informants reported that the "key player" did not physically assess the patient before deciding to proceed with the stroke alert, particularly in cases where revascularization therapy might be indicated.

"*The more skilled the on-call neurologist and the more obvious the stroke symptoms, the greater the possibility of managing the situation over the phone.*" *(Inf2)*

"*When there's suspicion of IHS, you contact the stroke coordinator, a nurse specialized in stroke care. They will ask you standardized questions based on a survey, and if the criteria are met, a stroke alert is initiated.*" *(Inf4)*

If the symptoms suggested a stroke, a stroke alert was initiated to notify radiology, and the patient was transported to the CT scan by the ward staff, where the physical examination would take place later. Informants emphasized that the "key player" could gather valuable information over the phone, guiding the caller through a screening evaluation if the symptoms were unclear. They underscored the crucial role of the "key player" in providing clear instructions to the ward staff regarding the urgency of the situation.

"*When you receive a phone call from someone who believes their patient had a stroke, especially from a ward you suspect is not accustomed to managing stroke patients, it's crucial to be explicit: "This is really urgent. Drop everything you're doing, bring a nurse, and meet me as soon as possible at the CT scan".*" *(Inf1)*

In cases where symptoms did not suggest stroke, or if revascularisation therapy was unlikely, the "key player" could examine the patient in the ward before making further recommendations. Some informants mentioned that the "key player" always assessed the patient in the ward before deciding to proceed with the stroke alert. In such cases, the "key player" played a role in expediting patient transport to radiology alongside the ward staff and initiated the physical examination that would otherwise occurr performed at the CT scan.

**The in-hospital stroke alert.** Most informants explained that their hospitals used the same infrastructure for IHS alerts as for COS alerts. The stroke alert for a COS typically consists of a page sent to all relevant staff (on-call neurologist/internist, radiologist, stroke nurse) and departments (radiology, emergency department, intensive care unit, stroke department). This alert is usually initiated by the emergency department's head nurse after receiving a call from paramedics regarding a suspected COS patient. Since the step where paramedics call the emergency department is absent in the care chain for IHS patients, informants described various ways to initiate the stroke alert from within the hospital.

The most common practice was for the "key player" to call the head nurse at the emergency department and request that they activate a stroke alert page. One informant described a system where the "key player" initiated the alert by calling the hospital switchboard, and the operator triggered the stroke alert via the pager system. This practice was established for all stroke alerts at that site. Another informant shared an alternative approach where the ward nurse was responsible for alerting relevant departments through phone calls, deviating from the standard practice for COS patients at that site. This highlights the variability in alert initiation procedures for IHS cases across different sites.

"...then we call the nurse at the main desk of the emergency department and inform them to go to the CT-scan [...] and they are supposed to activate the stroke alert by pushing a button, so radiology and ICU are alerted." (Inf5)

## 2. Obstacles to optimal practice

The second theme addresses obstacles to optimal IHS management. Examples include insufficient staff knowledge about when and how to act upon discovering an IHS patient, a weak general stroke alert system, and signs of resignation due to the perception that IHS is rare and management unlikely to improve.

**The unknown in-hospital stroke routine.** All but one site had a written IHS routine outlining how to initiate a stroke alert for a hospitalized patient with stroke symptoms. While some informants considered the IHS routines effective, conflicting information emerged during interviews. Some described doubt among staff about who to contact when suspecting stroke in a hospitalized patient. Others noted hesitancy in recognizing whether the symptoms indicated stroke and whether a stroke alert was indicated. Additionally, trust issues arose among the receivers of the page when the alert originated from a hospital ward rather than the emergency department. As a result, they concluded that stroke protocols designed for COS may not be universally recognized or trusted for IHS. Simplified protocols for when to suspect stroke in hospitalized patients and initiating the stroke alert were suggested, along with increased staff education to ensure adherence to existing routines.

"*The one thing you absolutely should not do is call the hospital switchboard and initiate the stroke alert. The stroke ward and radiology will not trust this stroke alert as it reaches them through a different channel. They are supposed to receive a phone call and a briefing; that is when they take action.*" (Inf2)

"*There is a stroke alert, it is initiated from the emergency department, but it was not established to be used from a ward.*" (Inf3)

At the hospital without an established routine, IHS management varied depending on the physician assuming the role of the "key player." While contacting the "key player" to initiate the stroke alert was common practice, subsequent steps in the care pathway often differed due to the absence of a standardized system known to all staff. However, the hospital's relatively small size allowed the "key player" to leverage local knowledge of clinics and personnel to devise tailored solutions for each IHS case.

"*We know most phone numbers by heart, I know my colleagues by their first name and recognize their voices on the phone. Therefore, it is possible to "reinvent the wheel" for each IHS case. But of course it is negative that we do not have an established routine.*" (Inf3)

**Lack of confidence.** Some informants described hesitancy among staff to initiate a stroke alert for a hospitalized patient, fearing they might be wrong about the symptoms indicating a stroke.

"*Then there's this threshold to initiate a stroke alert in a fairly well patient. So you find yourself thinking, "He is 82 years old and was quite tired yesterday too. Is there truly something happening or will someone just think that I'm embarrassing?*" (Inf5)

They noted that staff tended to seek confirmation from colleagues before contacting the "key player", which could lead to multiple evaluations within the ward before the stroke alert was initiated by calling the "key player". This hesitation was attributed to a lack of experience in managing stroke among those discovering the patient with suspected IHS.

> "*A nurse assistant informs the nurse about a suspected IHS, who then checks on the patient and feels uncomfortable because the doctor is at lunch. Should we disturb because of this? Nevertheless, you inform the doctor, who should recognize the urgency of the situation. Subsequently, the doctor should see the patient and understand that this requires immediate alerting. . . These steps before initiating the stroke alert are the most challenging.*" (Inf5)

> "*For wards other than stroke wards, it is crucial that the person who identifies a patient with unusual symptoms dares to act fast. By this, I mean calling the doctor immediately without thinking twice.*" (Inf6)

> "*The first steps are truly the hardest ones–to discover the patient, trust yourself, and dare to pursue the case.*" (Inf5)

This contrasts with COS, where the decision to activate the stroke alert is made in the pre-hospital setting. Ambulance staff, who manage patients with suspected stroke more frequently than hospital ward staff, are accustomed to using stroke screening protocols and benefit from the transport time to the hospital, which allows for a structured initial evaluation. If deemed necessary, they can contact the "key player" to discuss whether a stroke alert should be activated. Consequently, upon arrival at the hospital, the patient can proceed directly from the emergency department to the CT scan, as all relevant parties have had time to prepare in advance. Some informants emphasized that *maintaining* a high pace of management is easier than being the one responsible for *escalating* urgency, as in the case when ward staff must decide to call a stroke alert for a hospitalized patient without certainty about the symptoms. Simple protocols for when to call the "key player" when stroke is suspected in a hospitalized patient were suggested to encourage prompt action by staff.

> "*We are all in the right mindset in the emergency department, and the appropriate staff are present. Therefore, you can simply rush. . .*" (Inf2)

**A robust general stroke alert facilitates IHS management.** All informants described the radiology department as the natural meeting point for all involved parties, emphasizing that patient management for IHS should only deviate from COS management before arriving at the CT scan.

However, some informants highlighted challenges in IHS management that occurred after CT arrival, overlapping with the general stroke alert. Issues included suboptimal preparation for thrombolytic treatment (e.g., absence of IV lines, lack of weight information), delays at the CT-scan (routine CT-angiography preceding thrombolytics), and post-CT administration of thrombolytics in locations other than at the CT scan. Informants also noted staffing challenges during off-hours and perceived IHS patients as more medically complex.

> "*. . .then we move the patient from the CT-scan to a nearby room to administer the thrombolytic agent. This final step has proven to cause a delay of a minute or so, and we are exploring alternative approaches for the future.*" (Inf2)

Informants mentioned that revisions to their general stroke alert protocols positively impacted patient management, including IHS patients. These revisions included

administering thrombolytics before CT angiography, preparing thrombolytic agents before patient arrival, and increasing the level of medical competence at stroke alerts.

> *"During off-hours, stroke alerts used to be managed by interns. They were insecure and uncomfortable handling acute stroke patients. Now we only have residents or attendings working off-hours, and given their years of experience there is usually little to discuss." (Inf4)*

**Resignation.**   Some informants found IHS management acceptable despite its challenges, acknowledging the difficulty of eliminating the chaotic environment surrounding IHS cases in wards with no stroke experience. They suggested that educating staff about something rare might not be worthwhile, as the individual discovering the patient would still likely be inexperienced in stroke management. Some considered the existing routines for initiating the IHS alert adequate, attributing any issues to staff non-adherence. One informant felt that IHS management was as effective as COS management, although this was contradicted by background data.

> *"How good could it possibly be when there is always someone inexperienced involved. It will never be any better than expected, it will never be good. It could be as good as expected, and I believe that is how good it is now." (Inf2)*

> *"However, the management of IHS patients is no less effective than for those with community-onset." (Inf4)*

### 3. Knowledge and experience

The third theme addresses insufficient knowledge and experience with IHS as a factor influencing patient management, highlighting the pivotal role of education about IHS and hospital routines in increasing staff confidence to act.

**Varying levels of knowledge about HIS.**   All informants highlighted that there was insufficient education due to excessive administrative burdens and staff shortages. As a result, recent advances in acute stroke management, such as the possibility of performing thrombectomy within 24 hours, might not have been communicated to all staff.

> *"I definitely think that knowledge about this [thrombectomy within 24 h] is lagging behind, even among colleagues in internal medicine." (Inf3)*

Informants concurred that hospital staff lacked sufficient education on IHS, pointing out the absence of IHS-specific content in current education programs. They suggested that IHS education might be given a low priority due to the perception that it is rare, making the cost-benefit ratio seem unfavourable. Despite this, most informants agreed that regular education on IHS is essential to keep staff up to date.

> *"Of course, education of staff members at the wards could always be improved. But it is very resource-intensive, especially in a large hospital." (Inf1)*

> *"Especially those who do not work with stroke patients daily could use a reminder. It is easy to forget things you do not regularly encounter." (Inf6)*

**Knowledge and experience improve IHS management.**   The importance of knowledge and experience in stroke management was emphasized as crucial for successful IHS

management. Wards with prior experience in stroke management were reported to handle suspected IHS cases more effectively compared to inexperienced wards. This was attributed to staff experience in recognizing stroke symptoms, understanding the urgency of treatment, and familiarity with stroke management protocols.

"*When there is a stroke alert from the stroke ward, the staff naturally assume their positions. There is no need to assemble a separate stroke alert response team, you are the stroke alert response team.*" (Inf2)

"*You respond more easily to things you have heard about [IHS]. Then you have a strategy and a plan. Everything moves a bit faster when you are confident that this ward has a routine; you simply follow it and execute as instructed.*" (Inf5)

In contrast, wards with limited experience in stroke management were seen as more prone to delays in recognizing symptoms and initiating the stroke alert by calling the "key player." This was attributed to insecurity about interpreting stroke symptoms and unfamiliarity with the IHS routine. The rotation of interns to hospital wards with less stroke experience was considered a positive factor for enhancing IHS management, as their recent experience from other wards could contribute to improved recognition of stroke symptoms and timely initiation of the stroke alert.

"*I know our interns are very competent. As long as they are in the wards, [. . .] I'm convinced they will identify the patients with IHS.*" (Inf3)

**Education is requested.**   Several informants highlighted that staff were interested in stroke education, noting regular participation in stroke education sessions. Wards that had recently managed IHS patients expressed an interest in further improvement and actively sought additional IHS education. Constructive events were also described, including debriefing sessions held shortly after managing an IHS case. These sessions encouraged discussions of specific cases and emphasized the importance of following the IHS routine.

"*After a recent IHS case, we decided to designate the IHS alert as the "routine of the month". . . This way, we reminded each other about the routine, discussed it in workplace meetings and ensured everyone read it at least once.*" (Inf6)

Informants expressed a preference for brief, concise IHS education that focuses on when to suspect stroke in hospitalized patients and provides clear guidelines on the necessary actions. This approach aligns with the education provided for other urgent conditions, such as cardiopulmonary resuscitation.

"*In the best case scenario, calling an IHS alert should be common knowledge, just like I know how to manage urgent and dangerous conditions by heart, including some I've never seen, like anaphylactic shock [. . .] Shouldn't knowing how to initiate an IHS alert be a standard skill? Yes, I would like that!*" (Inf2)

## Discussion

We interviewed six physicians responsible for stroke alerts at different Swedish hospitals to explore IHS management. Based on these interviews, we summarized a schematic workflow

explaining the process for initiating a stroke alert in hospitalized patients, emphasizing the role of the "key player"—an individual with stroke expertise who has the authority to decide whether to proceed with immediate radiology or to abort the stroke alert. Informants highlighted that insufficient knowledge about stroke and IHS routines among hospital staff serves as a barrier to optimal management. This includes hesitation to contact the "key player" due to hesitancy about whether the patient's symptoms indicate a stroke. Suggestions for improvement included implementing simple protocols for contacting the "key player" and increasing staff knowledge of stroke in general, and IHS specifically, through education.

A key finding of this study was the identification of the "key player" as the primary contact in cases of suspected stroke symptoms in hospitalized patients. Contacting the "key player" initiated the stroke alert process, with this individual acting as a filter to determine whether to proceed with or abort the stroke alert. This decision was made either by gathering information over the phone or by conducting a rapid assessment of the patient on the ward. Subsequent alerts were only issued to radiology, the ICU, and the emergency department if the "key player" opted to continue with the stroke alert. The "key player" serves a role similar to the stroke response team described in the American Heart Association (AHA) scientific statement on IHS, particularly in confirming initial stroke symptoms and assessing the patient. However, a key distinction between our findings and the AHA's recommendations is the AHA emphasizes that the charge nurse is responsible for notifying radiology and pharmacology, whereas in our study, no other units were notified until the "key player" made the decision to proceed and activated their pagers. Given that the AHA reports nearly half of all in-hospital stroke alerts are for stroke mimics, the filtering role of the "key player" may help reduce unnecessary alerts. This, however, needs to be further studied.

We observed variations in the workflow, from patient identification to initiating an in-hospital stroke alert, across different sites. One notable difference was whether the "key player" made decisions regarding further management over the phone or after an in-person evaluation. Another variation was in how the stroke alert was activated: while some sites used preexisting infrastructure designed for COS patients, others had separate mechanisms for IHS alert activation. A potential explanation for these differences is the decentralized nature of healthcare in Sweden, which is managed by 21 different counties. In the absence of a national guideline for IHS management, each county—and in some cases, individual hospitals—develops its own protocols for managing stroke in hospitalized patients. Despite these variations, the workflows across sites were relatively similar within this context. However, it remains unclear whether any of the described strategies is superior in ensuring rapid treatment delivery.

Several obstacles were identified in the study. One challenge was that ward staff were unfamiliar with the IHS routine and therefore, despite having access to written guidelines on IHS, were hesitant to contact the "key player". This stemmed from hesitancy about whether the patient's symptoms were stroke-related and whether a stroke alert was warranted. Additionally, some staff were unsure who to contact in the event of suspected stroke symptoms in a hospitalized patient. These issues sometimes led to discussions among ward staff and multiple patient evaluations before ultimately deciding to activate the stroke alert by contacting the "key player". Although IHS is relatively common, it is unlikely for a staff member in a given hospital to encounter a patient developing a stroke during their shift. Most staff outside the stroke ward or emergency department lack direct experience with stroke unless they have previous experience in those areas. Therefore, it is crucial to establish systems that support and encourage immediate action by the individual who identifies a suspected IHS case.

For example, many wards in Sweden are equipped with a "code blue" button that staff can press to initiate an immediate response from ICU and cardiology teams. The only criteria for activation are suspected cardiac or respiratory arrest. Although the button is sometimes

pressed for non-cardiac arrest emergencies, it signals a life-threatening event requiring urgent intervention. In comparison, in-hospital stroke is a more heterogeneous condition, presenting with a wider range of symptoms from hemiplegia to aphasia, neglect, or vertigo. Creating a universal protocol for stroke alerts initiated by pressing a button may be challenging due to the risk of both over- or under-diagnosis. A more pragmatic approach may be to implement a simple protocol encouraging staff to contact the "key player" if a patient develops any stroke symptoms within the last 24 hours. The "key player" could then decide whether to proceed with the stroke alert or to abort it. In this way, the patient will still receive appropriate management, but it does not necessarily need to occur within the formal stroke alert care chain and most importantly: the decision about proceeding with the stroke alert is placed on the "key player", removing the aspect of hesitation from the staff, possibly mitigating the delay in calling. Whether such a protocol, as suggested during the interviews, would amend the hesitancy reported remains to be studied.

Some obstacles were related to steps in the stroke alert process occurring after the patient's arrival for CT, affecting both IHS and COS patients. However, the medical complexity of IHS patients emerged as a distinct factor complicating treatment decisions, indicating a need for more experienced physicians to be involved in the stroke alert process for hospitalized patients. Some informants expressed a sense of resignation, believing that IHS management is on par with COS management, that further improvement may not be achievable, or that IHS is too rare to warrant additional effort in optimizing its management. The perceptions about IHS management being on par with COS management are contradicted by data showing that IHS patients experience twice the time to intravenous thrombolysis compared to COS patients. Given the critical importance of timely treatment in stroke outcomes, every opportunity to reduce delays should be explored. For IHS patients, the goal should be to achieve a DNT similar to that of COS patients. In-hospital cardiac arrest, which accounts for approximately 30% of all cardiac arrests [23], benefits from well-established management protocols due to the urgency of treatment [24]. While IHS is two to three times less common than in-hospital cardiac arrest in absolute numbers, its time-sensitive nature similarly warrants the development of robust management routines.

Several strategies were suggested to address the identified obstacles. One recommendation was to increase education to improve general knowledge about stroke and stroke treatment while ensuring that all staff are familiar with the IHS routine and the criteria for contacting the "key player". Given the global efforts to raise public awareness about stroke, it is reasonable to expect all healthcare professionals to possess a basic understanding of stroke management. In today's digitalized world, it is feasible to provide short yet informative texts or videos to a large audience, adressing concerns that education is resource-intensive or not worth the effort. Debriefing sessions were also proposed, where staff could engage in constructive discussions about IHS management and explore areas for improvement. While our data do not explicitly support this, it seems logical to assume that having an IHS routine in place is preferable to having none.

Given the limited sample size of this study, it is inappropriate to draw broad conclusions or make specific recommendations based on our findings. Future studies in Sweden could investigate workflows and barriers to optimal IHS management across more hospitals, potentially by using standardized questionnaires. This would allow for analysis of the association between management strategies, hospital characteristics, and the efficiency of IHS management, as well as patient outcomes. Experimental studies investigating the effects of targeted interventions are also needed to elucidate the optimal workflow for IHS.

### Methodological strengths, limitations and reflexivity

The strengths of this study include a purposive sampling method, which provided diverse perspectives from Swedish hospitals with both fast and slow IHS lead times. The study's specific aim and the relevant expertise of the informants enriched the discussions and ensured high information power despite the limited sample size [21]. However, the workflows, strategies, and perceptions of IHS at sites not included in the study may differ, making the transferability of the results to other hospitals within Sweden or internationally uncertain. One limitation was that we did not share preliminary results with participants, which might have provided an opportunity to explore additional aspects. Additionally, the study was limited by the exclusion of hospital staff other than physicians. We reflected on our roles as researchers in the knowledge production process in line with RTA principles [22], acknowledging that high-quality RTA involves actively reflecting on our biases during data collection and interpretation. The diverse backgrounds of the authors further enhanced the discussions and analytical process.

## Implications

This study identified variations in workflows for in-hospital stroke (IHS) management across the six sites, while also highlighting a common factor at all sites: the presence of a "key player," who served as the primary contact for initiating the stroke alert. Clear guidelines for contacting the "key player" and enhanced education on IHS were suggested as potential strategies to reduce delays in activating the stroke alert. Future studies could involve a larger sample of hospitals and investigate whether specific management strategies or hospital factors are associated with faster IHS management and treatment.

## Supporting information

**S1 Table. Aggregated lead time data.**
(DOCX)

**S1 Text. Transcript informant 1 (English).**
(DOCX)

**S2 Text. Transcript informant 2 (English).**
(DOCX)

**S3 Text. Transcript informant 3 (English).**
(DOCX)

**S4 Text. Transcript informant 4 (English).**
(DOCX)

**S5 Text. Transcript informant 5 (English).**
(DOCX)

**S6 Text. Transcript informant 6 (English).**
(DOCX)

## Acknowledgments

We thank the informants for participating.

## Author Contributions

**Conceptualization:** Ilan Ben-Shabat, Kristina Lindvall, Jonatan Salzer.

**Data curation:** Ilan Ben-Shabat, Kristina Lindvall, Jonatan Salzer.

**Formal analysis:** Ilan Ben-Shabat, Kristina Lindvall, Jonatan Salzer.

**Funding acquisition:** Ilan Ben-Shabat, Jonatan Salzer.

**Investigation:** Ilan Ben-Shabat, Kristina Lindvall, Jonatan Salzer.

**Methodology:** Ilan Ben-Shabat, Kristina Lindvall, Jonatan Salzer.

**Project administration:** Ilan Ben-Shabat.

**Supervision:** Kristina Lindvall, Jonatan Salzer.

**Writing – original draft:** Ilan Ben-Shabat, Kristina Lindvall, Jonatan Salzer.

**Writing – review & editing:** Ilan Ben-Shabat, Kristina Lindvall, Jonatan Salzer.

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
