## [Decision Letter · Decision Letter 0]

30 May 2024

PONE-D-24-16819Exploring strategies for management of in-hospital stroke in Sweden: a qualitative studyPLOS ONE

Dear Dr. Ben-Shabat,

Thank you for submitting your manuscript to PLOS ONE. After careful consideration, we feel that it has merit but does not fully meet PLOS ONE’s publication criteria as it currently stands. Therefore, we invite you to submit a revised version of the manuscript that addresses the points raised during the review process.

We look forward to receiving your revised manuscript.

Kind regards,

Sonu Bhaskar, MD PhD FANA

Academic Editor

PLOS ONE

Journal Requirements:

 "IB, 250.000 SEK, Region Norrbotten, https://www.norrbotten.se

JS, 100.000 SEK, VISARE NORR Fund, Northern Country Councils Regional Federation, https://www.norrasjukvardsregionforbundet.se/halso-och-sjukvard/fou/visare-norr/

This study was conducted without involvement of the funders."

4. In the online submission form, you indicated that [The transcripts cannot be shared publicly without limitations because it could be possible to identify the participants even though we pseudonymize the data. Data are available upon request from IB for researchers who meet the criteria for access to confidential data.]. 

Additional Editor Comments:

Thank you for submitting your work to PLOS One. Your work has now been reviewed by two independent experts. Based on the feedback from the reviewers and a careful review of the work, I would like to invite you to revise your manuscript and make appropriate changes as per the comments/feedback provided below. Please submit your revised manuscript with a point-by-point rebuttal.

Additionally, there are some additional points to consider:

1. Introduction:

The introduction can be expanded to highlight previous work on in-hospital stroke as well as the role of systems interventions in improving stroke workflows and outcomes, especially in patients receiving reperfusion therapy. For example, Ackah et al. have provided insights on 30-day in-hospital case fatality and risk factors from Sub-Saharan Africa (see Ackah et al., PLOS Glob Public Health. 2024; 4(1): e0002769).

With regards to systems interventions in reducing delays in acute stroke treatment, including in-hospital interventions, we suggest discussing findings from the study by Baskar PS et al. (Acta Neurol Scand 2021; PMID: 32882056 and PMID: 34101170).

2. Inclusion and Exclusion Criteria:

Please provide details on the inclusion and exclusion criteria, if applicable.

3. Limitations:

In the limitations section, it may be discussed that workflows applicable to this center may be different from other centers and hence may influence the management strategies and perceptions.

Reviewers' comments:

Reviewer's Responses to Questions

**Comments to the Author**

1. Is the manuscript technically sound, and do the data support the conclusions?

Reviewer #1: No

Reviewer #2: Yes

2. Has the statistical analysis been performed appropriately and rigorously? 

Reviewer #1: Yes

Reviewer #2: N/A

3. Have the authors made all data underlying the findings in their manuscript fully available?

Reviewer #1: Yes

Reviewer #2: No

4. Is the manuscript presented in an intelligible fashion and written in standard English?

Reviewer #1: Yes

Reviewer #2: No

5. Review Comments to the Author

Reviewer #1: Thank you for the opportunity to review this qualitative study on in-hospital stroke in Sweden. Reading that the interviews were done 2 years ago, I suspect that this article has been submitted to multiple journals with limited success. I actually think the authors have something worth publishing, but the article needs significant revisions. Therefore, I spent a lot of time trying to help the authors understand a viewpoint that may be the cause for prior rejections. Namely – this does not follow traditional qualitative research methodology and they are not adequately addressing information that has already been published elsewhere. I hope that inclusion of this information in a future revision will help lead them towards a successful publication. My comments are divided as MAJOR (must be addressed) and MINOR (not required, but worthy of consideration by the author).

Major Considerations

The purpose of qualitative research is theory generating … and if you are drawing conclusions from qualitative research you are over-reaching the purpose of that methodology. The “conclusion” from qualitative research are typically much more about “implications” and may identify potential hypotheses which are yet to be tested.

Qualitative research is not appropriate to detail workflows. Qualitative methodology is appropriate to provide a deeper understanding of a phenomenon. I think it would be accurate to write “

The subtheme “tentative path to an in-hospital stroke alert” reads much more like “Hierarchy” or “process barriers.” This resonates with previous research (both qualitative and quantitative) on stroke code activation (both IHS and community onset). I would ask that you carefully consider either renaming this, or clearly identify the similarities and differences between your theme and previously established constructs.

The subtheme “tentative path to an in-hospital stroke alert” which I would argue is identified in literature as “Hierarchy” or “process barrier” seems like it would fit better in the second major theme of OBSTACLES. – you don’t have to move it, but you should defend why you are not placing this as an obstacle when it is previously identified as such.

Page 10 – middle. When you talk about “trust issues” you don’t have any quotes or data to back that up? Also, trust, team, parallel processing, and communication are well-established in the literature as vital to the success of stroke code for community onset. And have early established links for IHS.

Informant 3 brought out specifically that there is a lack of a standardized routine. This is echoed across several other comments in regard “different” processes being described. The failure to have an established routine is also clearly identified (including by AHA, ESO, and GWTG-Stroke). First and foremost – how was this NOT a theme? But if you are not going to go back and look at your coding – you really need to address how and why your sample has such diversity compared to existing literature.

Page 11 – who attributed “hesitancy and uncertainty” to a lack of experience? Was it the 3 authors or was it persons you interviewed? These constructs have also been linked lack of standardized processes and hierarchical models of care. As you point out in the bottom paragraph (same page) the staff who see a community-onset stroke are not hesitant and not uncertain . . . so knowledge and experience do not appear to be the barrier. Bottom of page 12 further solidifies that changes to stroke protocol made a difference and this further nullifies the link between hesitation and experience. You might look up “confidence” and barriers to confidence as process factors are more directly linked to confidence than to experience.

Theme 3 is titled “Education and knowledge” and then immediately described as “knowledge and experience” then later in the sentence as “education and experience” but experience does not equal education. This is confusion and I recommend that you either edit the sentence or redefine the theme.

Page 14 – informant 1 states “resources” and resources have been identified in early literature as a barrier to implementation of best practice for stroke treatment. This is not new. I would ask that you include references to pre-existing literature.

Page 16 – second sentence of discussion goes against international recommendations and if your recommendation is to have a KEY PLAYER initiate the stroke alert instead of empowering all staff to initiate a stroke alert – you much vigorously defend this position and provide new evidence to support this change in practice and critically evaluate the existing literature supporting empowerment of all humans to initiate the chain of survival.

I stopped providing recommendations at discussion section because that will be entirely different after editing based on the above.

The following are some additional references. This is by no means an exhaustive list.

• PMID: 19901755

• PMID: 24135928

• PMID: 31104615

• PMID: 32168017

• PMID: 33691469

• PMID: 35123276

• PMID: 35522791

• PMID: 36383064

• PMID: 36748462

• PMID: 36749865

• PMID: 37103061

• PMID: 37502264

• PMID: 37549479

• PMID: 37556461

• PMID: 37732490

Minor Considerations

Avoid writing paragraphs that are 1 or sentences long. Write complete paragraphs. For example, the introduction could be written as one well-written paragraph instead of 3 incomplete paragraphs.

The more modern writing style does not use so many subheaders (that is a holdover from APA). For example, the methods section has 6 headers for what could be 3 well written paragraphs.

Please place tables at the end of the submission and after references. Cite the table in the body, but the table itself goes at the end.

Qualitative methodology would describe the sampling method as “purposive sampling.” To be more precise, I would add this language. But I do compliment how thoroughly you explain the process.

Requiring the nurse or on-site physician to check with an attending physician (KEY PLAYER) before calling a stroke code is bizarre. If the nurse was at a restaurant and saw a person become aphasic with hemiparesis – would you expect them to call a physician before activating the chain of survival? If a nurse witnessed a patient with cardiac arrest in the hospital – do you ask them to check with the ward physician – who then calls the attending at home - before they begin resuscitation efforts? This does not read as though the practice in your community is up to international standards. Later there is a “stroke coordinator” who is empowered to activate a stroke code . . . again – this doesn’t make sense operationally and is worthy of discussion.

Page 8 2nd paragraph in Key Player section. Change to read “When the key player is notified . . .” instead of “Once the key player…” (jargon and imprecise).

On the top of Page 10 the word in the 2nd line is “staff” not “stauff”

“Uncertainty” is a well-described phenomenon in care provision and as such you need to be explicit about how you are using this term.

In the section on RESIGNATION you argue that some feel negatively toward educating about a rare event. Yet, there are world-wide efforts to educate the general-population about stroke. This would be an important place to insert and discuss relevant related research. Or perhaps argue that all these international education efforts are a complete waste of time.

The informant 4 statement that “… management of IHS is not less effective than community-onset stroke..” would benefit from contrasting data (some of which is in your own background) that contradicts this nihilistic statement.

Do not use and/or in professional writing (or at least almost never). Most often, and/or is and; sometimes and/or is or. Example page

Reviewer #2: Thank you for giving me a chance to review this paper. This study focused on an important health issue but there are few comments that should be addressed by authors as listed below:. I am listing some below:

1. Please write the abstract based on the journal guideline.

2. How did you find 6 participants are enough for this study?

3. How did you develop the study tool? How many sections had? Please attach the questionnaire.

4. Please explain how did you recruit study participants? How did you contact them? Who was involved on it?

5. How did you conduct interviews? In what languages? How about translating the content of interview? Who has translated it to English?

6. Was there any relationship between the first author and 6 study participants?

7. Please write the study credibility, transferability, dependability,… as study trustworthiness under the “study rigor” in the methodology section.

8. please add the 32 items COREQ (COnsolidated criteria for REporting Qualitative research) Checklist.

9. please add a conclusion after the Discussion and provide some recommendations.

6. PLOS authors have the option to publish the peer review history of their article (what does this mean?). If published, this will include your full peer review and any attached files.

Reviewer #1: No

Reviewer #2: **Yes: **Masoud Mohammadnezhad

---

## [Author Response · Author response to Decision Letter 0]

23 Oct 2024

Dear Editor and Reviewers,

Thank you for the opportunity to revise our manuscript and for your thoughtful and constructive comments. We greatly appreciate the time and effort you have taken to review our work. Your insights have been invaluable in helping us improve the manuscript, and we have carefully addressed each of your suggestions in the revised version. We hope these revisions meet your expectations and further enhance the quality of the study.

Journal Requirements

We adjusted the title page, heading style and table/figure citing to match the journals style requirements.

Thank you for pointing this out! The new statement is included in the cover letter (the correct statement is the one in the next comment).

 "IB, 250.000 SEK, Region Norrbotten, https://www.norrbotten.se

JS, 100.000 SEK, VISARE NORR Fund, Northern Country Councils Regional Federation, https://www.norrasjukvardsregionforbundet.se/halso-och-sjukvard/fou/visare-norr/

This study was conducted without involvement of the funders."

Thank you for pointing this out. Since the funders had no role in the study, we included this statement to the cover letter.

4. In the online submission form, you indicated that [The transcripts cannot be shared publicly without limitations because it could be possible to identify the participants even though we pseudonymize the data. Data are available upon request from IB for researchers who meet the criteria for access to confidential data.]. 

Thank you for this information. After reviewing our ethical approval, we found no restrictions preventing us from sharing our results. Since the informants were interviewed in their professional roles, we determined the information to be less sensitive compared to patient data. Consequently, we used a GPT-4 model (ChatGPT, OpenAI) to produce a verbatim translation of the pseudonymized interviews from Swedish to English and added them as Supporting information together with our manuscript.

Additional Editor Comments

1. Introduction:

The introduction can be expanded to highlight previous work on in-hospital stroke as well as the role of systems interventions in improving stroke workflows and outcomes, especially in patients receiving reperfusion therapy. For example, Ackah et al. have provided insights on 30-day in-hospital case fatality and risk factors from Sub-Saharan Africa (see Ackah et al., PLOS Glob Public Health. 2024; 4(1): e0002769).

With regards to systems interventions in reducing delays in acute stroke treatment, including in-hospital interventions, we suggest discussing findings from the study by Baskar PS et al. (Acta Neurol Scand 2021; PMID: 32882056 and PMID: 34101170).

Thank you for this comment! We expanded the introduction with information about the effects of systems interventions in the stroke care chain as well as prior studies on interventions on the IHS care chain, where the latter also included previously identified barriers (page 3, “Background”).

2. Inclusion and Exclusion Criteria:

Please provide details on the inclusion and exclusion criteria, if applicable.

The description about informant selection can be found on page 4 (“Methods - Design, settings and informants”).

3. Limitations:

In the limitations section, it may be discussed that workflows applicable to this center may be different from other centers and hence may influence the management strategies and perceptions.

We agree and expanded the section about transferability, discussing that our results from a relatively small sample study (6 sites) might not be generalizable to other hospitals (page 21, “Methodological strengths, limitation and reflexivity”).

Reviewers' comments

Reviewer #1

Thank you for the opportunity to review this qualitative study on in-hospital stroke in Sweden. Reading that the interviews were done 2 years ago, I suspect that this article has been submitted to multiple journals with limited success. I actually think the authors have something worth publishing, but the article needs significant revisions. Therefore, I spent a lot of time trying to help the authors understand a viewpoint that may be the cause for prior rejections. Namely – this does not follow traditional qualitative research methodology and they are not adequately addressing information that has already been published elsewhere. I hope that inclusion of this information in a future revision will help lead them towards a successful publication. My comments are divided as MAJOR (must be addressed) and MINOR (not required, but worthy of consideration by the author).

Major Considerations

1. The purpose of qualitative research is theory generating … and if you are drawing conclusions from qualitative research you are over-reaching the purpose of that methodology. The “conclusion” from qualitative research are typically much more about “implications” and may identify potential hypotheses which are yet to be tested.

Thank you for this comment. We rephrased and renamed the conclusions paragraph on page 21-22 to “Implications”. We now relate to the workflows that were identified and suggest further research to learn more about the subject.

2. Qualitative research is not appropriate to detail workflows. Qualitative methodology is appropriate to provide a deeper understanding of a phenomenon. I think it would be accurate to write…

Thank you for highlighting this point. Our study aimed not only to describe the IHS workflow but also to gain a deeper understanding of the factors that may influence it. This broader objective is what led us to select a qualitative methodology. We have clarified our aims accordingly in the final paragraph of the introduction (page 3).

3. The subtheme “tentative path to an in-hospital stroke alert” reads much more like “Hierarchy” or “process barriers.” This resonates with previous research (both qualitative and quantitative) on stroke code activation (both IHS and community onset). I would ask that you carefully consider either renaming this, or clearly identify the similarities and differences between your theme and previously established constructs.

We agree with the reviewer comment and renamed the subtheme to “Lack of confidence”.

4. The subtheme “tentative path to an in-hospital stroke alert” which I would argue is identified in literature as “Hierarchy” or “process barrier” seems like it would fit better in the second major theme of OBSTACLES. – you don’t have to move it, but you should defend why you are not placing this as an obstacle when it is previously identified as such.

We agree with the reviewer comment and moved the subtheme to the second major theme: Obstacles to optimal practice (page 13).

5. Page 10 – middle. When you talk about “trust issues” you don’t have any quotes or data to back that up? Also, trust, team, parallel processing, and communication are well-established in the literature as vital to the success of stroke code for community onset. And have early established links for IHS.

Thank you for this comment! There is a quote about trust from Inf2 (page 12, “The unknown in-hospital stroke routine”). 

Correct, we therefore added a section in the introduction about system interventions for COS and prior interventions on the care chain of IHS to highlight these aspects (page 3, “Introduction”).

6. Informant 3 brought out specifically that there is a lack of a standardized routine. This is echoed across several other comments in regard “different” processes being described. The failure to have an established routine is also clearly identified (including by AHA, ESO, and GWTG-Stroke). First and foremost – how was this NOT a theme? But if you are not going to go back and look at your coding – you really need to address how and why your sample has such diversity compared to existing literature.

One out of six sites did not have a written routine for IHS management which is mentioned in the second major theme (page 12, “the unknown IHS routine”). From the interviews it appeared that even though the routine was there, hospital staff had issues with using it stemming from unclarities within the routine on how to identify a stroke and who to contact. 

In Sweden, there is no national IHS guideline. We have added information on this in the introduction (page 3, last paragraph) and further discussed what effects this may have in the discussion (page 19, middle paragraph).

7. Page 11 – who attributed “hesitancy and uncertainty” to a lack of experience? Was it the 3 authors or was it persons you interviewed? These constructs have also been linked lack of standardized processes and hierarchical models of care. As you point out in the bottom paragraph (same page) the staff who see a community-onset stroke are not hesitant and not uncertain . . . so knowledge and experience do not appear to be the barrier. Bottom of page 12 further solidifies that changes to stroke protocol made a difference and this further nullifies the link between hesitation and experience. You might look up “confidence” and barriers to confidence as process factors are more directly linked to confidence than to experience.

Thank you for this valuable comment. In the interviews, it was noted that staff inexperienced with stroke might hesitate both with evaluating stroke symtoms and managing stroke patients. When referring to staff interacting with COS patients, we are primarily referring to ambulance personnel, who are generally well-experienced with stroke and therefore more confident in their management and evaluation. We agree that confidence plays a significant role in overcoming inexperience, particularly through the implementation of standardized protocols that encourage action, even among less experienced staff. This was also mentioned in the interviews. In response to your comment, we have added a section in the results (page 14, “Lack of confidence”) and the discussion (pages 19-20, “Discussion”) addressing this point.

8. Theme 3 is titled “Education and knowledge” and then immediately described as “knowledge and experience” then later in the sentence as “education and experience” but experience does not equal education. This is confusion and I recommend that you either edit the sentence or redefine the theme.

Thank you for pointing this out. We changed the name of theme to “Knowledge and experience” (page 15, “Knowledge and experience”).

9. Page 14 – informant 1 states “resources” and resources have been identified in early literature as a barrier to implementation of best practice for stroke treatment. This is not new. I would ask that you include references to pre-existing literature.

We added a sentence about resources in the introduction as well as a reference (page 3, “Introduction”).

10. Page 16 – second sentence of discussion goes against international recommendations and if your recommendation is to have a KEY PLAYER initiate the stroke alert instead of empowering all staff to initiate a stroke alert – you much vigorously defend this position and provide new evidence to support this change in practice and critically evaluate the existing literature supporting empowerment of all humans to initiate the chain of survival.

Thank you pointing this out. We believe this may be a matter of when the stroke alert is deemed to be activated. What we previously described as “Stroke alert activation” is more appropriately described as the decision to perform immediate radiology. We have now clarified that calling the ”key player” is the first step when initiating the stroke alert. We also clarified that our study makes no recommendation but merely reports and discusses the findings (page 21, “Discussion”).

11. I stopped providing recommendations at discussion section because that will be entirely different after editing based on the above.

Thank you for your insightful comments. As anticipated, the discussion section has been completely rewritten based on the revisions and suggestions provided.

12. The following are some additional references. This is by no means an exhaustive list.

• PMID: 19901755

• PMID: 24135928

• PMID: 31104615

• PMID: 32168017

• PMID: 33691469

• PMID: 35123276

• PMID: 35522791

• PMID: 36383064

• PMID: 36748462

• PMID: 36749865

• PMID: 37103061

• PMID: 37502264

• PMID: 37549479

• PMID: 37556461

• PMID: 37732490

Thank you for providing this list of references. We highlighted those added to the manuscript.

Minor Considerations

13. Avoid writing paragraphs that are 1 or sentences long. Write complete paragraphs. For example, the introduction could be written as one well-written paragraph instead of 3 incomplete paragraphs.

Thank you for this comment. We have revised the manuscript to ensure that paragraphs are more complete and cohesive.

14. The more modern writing style does not use so many subheaders (that is a holdover from APA). For example, the methods section has 6 headers for what could be 3 well written paragraphs.

We have reduced the number of subheadings and combined related content into cohesive paragraphs.

15. Please place tables at the end of the submission and after references. Cite the table in the body, but the table itself goes at the end.

We have kept the table in the body according to the journal guidelines: “Tables should be included directly after the paragraph in which they are first cited”.

16. Qualitative methodology would describe the sampling method as “purposive sampling.” To be more precise, I would add this language. But I do compliment how thoroughly you explain the process.

Thank you for this compliment! We changed the description of the sampling method to “purposive sampling” according to comment (page 4, “Design, settings and informants”).

17. Requiring the nurse or on-site physician to check with an attending physician (KEY PLAYER) before calling a stroke code is bizarre. If the nurse was at a restaurant and saw a person become aphasic with hemiparesis – would you expect them to call a physician before activating the chain of survival? If a nurse witnessed a patient with cardiac arrest in the hospital – do you ask them to check with the ward physician – who then calls the attending at home - before they begin resuscitation efforts? This does not read as though the practice in your community is up to international standards. Later there is a “stroke coordinator

---

## [Editor Report · Decision Letter 1]

31 Oct 2024

Exploring strategies for management of in-hospital stroke in Sweden: a qualitative study

PONE-D-24-16819R1

Dear Dr. Ben-Shabat,

We’re pleased to inform you that your manuscript has been judged scientifically suitable for publication and will be formally accepted for publication once it meets all outstanding technical requirements.

Kind regards,

Sonu Bhaskar, MD PhD FANA

Academic Editor

PLOS ONE

Additional Editor Comments (optional):

Thank you for submitting the revised version of your manuscript. I'm pleased to accept the manuscript. Congratulations and thank you for submitting your work to PLOS One.
---

## [Editor Report · Acceptance letter]

15 Nov 2024

PONE-D-24-16819R1 

PLOS ONE

Dear Dr. Ben-Shabat, 

I'm pleased to inform you that your manuscript has been deemed suitable for publication in PLOS ONE. Congratulations! Your manuscript is now being handed over to our production team.

Kind regards, 

on behalf of

Dr. Sonu Bhaskar 

Academic Editor

PLOS ONE